

# Bone-tissue decomposition of a single X-ray image *via* solving a Laplace equation

Zhili Wei[1], Wenming Tang[1] and Yuanhao Gong[2]

[1] School of Intelligent Manufacturing, Shenzhen Institute of Information Technology, Shenzhen, China
[2] College of Electronics and Information Engineering, Shenzhen University, Shenzhen, China

## ABSTRACT

Because bones are often enveloped by soft tissues, their visibility in X-ray images is compromised, resulting in a lack of clarity. Addressing this challenge, our article introduces an innovative approach to virtually decompose an X-ray image into distinct components: one representing soft tissues and the other, the bone structure. To achieve this separation, we have formulated a novel mathematical model. With proper assumptions, the model is reduced to a standard Laplace equation, which has fast numerical solvers. Our method has two important properties. First, the bone image derived from this process is theoretically guaranteed to have enhanced contrast relative to the original, thereby accentuating the visibility of bony details. Second, our method is computationally fast. Our method can process a 2,044 × 1,514 resolution image within 0.35 s on a laptop (8.8 million pixels per second). Our methodology has been validated through a series of numerical experiments, demonstrating its efficacy and efficiency. With such performance, this technique holds promise for a broad spectrum of X-ray imaging applications, including but not limited to clinical diagnostics, surgical planning, pattern recognition, and advanced deep learning applications.

# INTRODUCTION

X-ray has been widely used in the field of biomedical imaging and clinical diagnosis, particularly in bone research and human body diagnosis (*Chapman et al., 1997*; *Sakdinawat & Attwood, 2010*; *Gong et al., 2019*; *Ou et al., 2021*; *Huang, Wu & Gong, 2023*; *Ataei et al., 2024*). Since its discovery and development in 1895, X-ray imaging technology has become immensely popular and is now extensively used in various research and industry domains. In present times, it has become an essential tool for diagnosing bone-related conditions in clinical applications.

The skeletal system, comprising of bones, acts as a central pillar in maintaining the structure of mammalian bodies. These bones provide a framework that supports the body and protects the organs they encase. The health, shape, and integrity of these bones play an incredibly vital role in the overall health, functionality, and well-being of an individual, enabling a wide array of activities such as walking, running, dancing, and even the simple task of standing upright. Due to the pivotal role bones play in overall health, bone studies

Corresponding author
Yuanhao Gong, gong.ai@qq.com

and research into their health, structure, and function have become a significant and rapidly growing area of research.

In mammals, bones are commonly wrapped and enshrined in a variety of soft tissues that provide both support and protection. These tissues not only help in the protection of bones but also supply the necessary nutrients and stimuli for bone growth, regeneration, and remodeling. Additionally, they act as a crucial barrier against external forces, shocks, and injuries, further enhancing the bone's ability to perform its functions effectively.

Bones can be distinguished from these surrounding soft tissues by various characteristics or factors such as their shape, thickness, hardness, and density-attributes that make them unique in their structure and function. However, when these bones are covered or obscured by soft tissue, it can pose a significant challenge for medical professionals to see the details on bones.

Notably, X-ray imaging, a modality utilizing electromagnetic radiation, enables the critical differentiation of bones and soft tissue structures. The fundamental physical basis of this technique lies in the differential attenuation of X-ray photons as they traverse anatomical structures. During radiography examination, incident X-rays penetrate the body and undergo varying degrees of absorption and scatter contingent upon the density and atomic composition of the intervening tissues. Bone structures, characterized by significantly higher density and effective atomic number compared to soft tissues, induce substantially greater attenuation of the incident radiation. Consequently, the radiation flux reaching the detector (*e.g.*, film or digital sensor) exhibits pronounced spatial variation. This differential attenuation is manifested in the resultant image as distinct contrast gradients, clearly delineating the boundaries and relative radio-densities of skeletal elements against the surrounding soft tissue matrix. Figure 1 provides two examples of this phenomenon in the left column.

## X-ray images

Dense structures like bones block more X-rays, making them appear darker on the sensor compared to other areas. On the other hand, less dense and thinner soft tissue allows more X-rays to pass through, resulting in brighter areas on the sensor.

To improve the visibility of bone regions, modern X-ray images often use a subtraction process. This involves subtracting a constant maximum value, caused by the X-ray dose, from the original image. This operation makes bone regions appear brighter and soft tissue regions appear darker, making the bone region more distinguishable.

In clinical applications, the window technique is commonly used to limit the intensity of bone within a specific range. This effectively removes soft tissue regions that do not overlap with bones. However, this method cannot remove soft tissue that overlaps with bone regions.

To achieve better visualization of bones, other methods aim to enhance image contrast through techniques like histogram equalization, Contrast Limited Adaptive Histogram Equalization (CLAHE), and more (*Gong et al., 2019*; *Aldoury et al., 2023*; *Yan et al., 2023*). However, these methods often enhance both bone and soft tissue simultaneously, resulting in visually improved images but introducing a complex or unknown relationship between

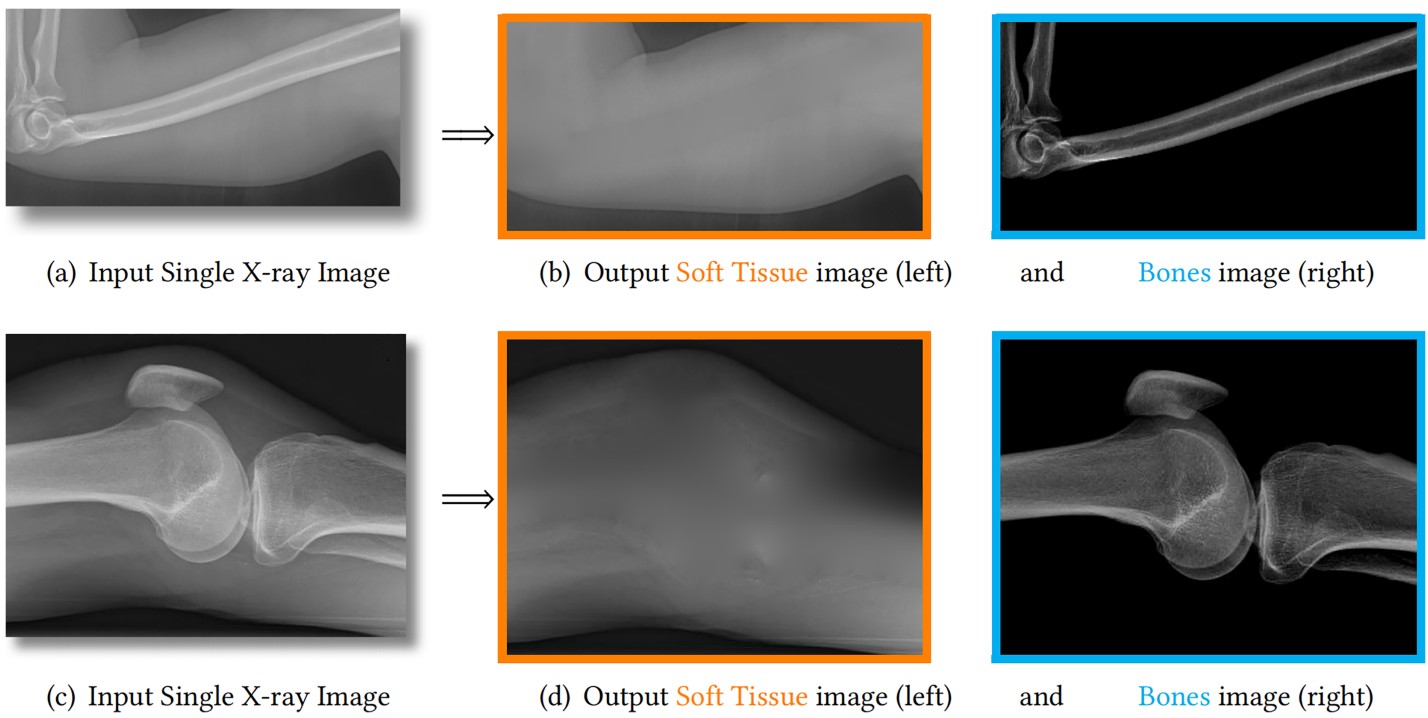

**Figure 1** **We propose to decompose one X-ray image (A, C) into one soft tissue image (B, D, orange) and one bone image (B, D, blue).** The estimated bones are theoretically guaranteed to have larger image contrast than the input image. Therefore, the details on bones are enhanced and becomes much clearer.

image intensity and the actual X-ray dose. Such a complex or unknown relationship causes artifacts and makes the results more difficult to interpret.

To address these challenges, this article proposes a method to separate an X-ray image into a soft tissue image and a bone image. This decomposition technique maintains the linear relationship between image intensity and the physical properties of the objects being imaged. Figure 1 showcases two examples of our method, demonstrating the guaranteed bone enhancement. Further details about our method will be explained in subsequent sections.

## Scattered light in physics

As shown in the left column of Fig. 1, bones are usually surrounded by soft tissue, which is similar to many natural scenes. One example is foggy weather, as depicted in Fig. 2A, where the fog can be considered as "soft tissue" (low density) and the buildings as "bone" (high density).

The phenomenon behind this is called light scattering (*Van de Hulst, 1958*), which occurs in different situations, including X-ray images in clinics, foggy weather in natural scenes, and fluorescence images in biological imaging (*Gong & Sbalzarini, 2016*; *Gong, 2015*). Scattered light can affect the quality of an image, such as when soft tissue scatters X-rays, making bone details less clear.

The study of light scattering dates back to 1871 when Rayleigh examined this phenomenon for light wavelengths larger than the particles' radius in the medium
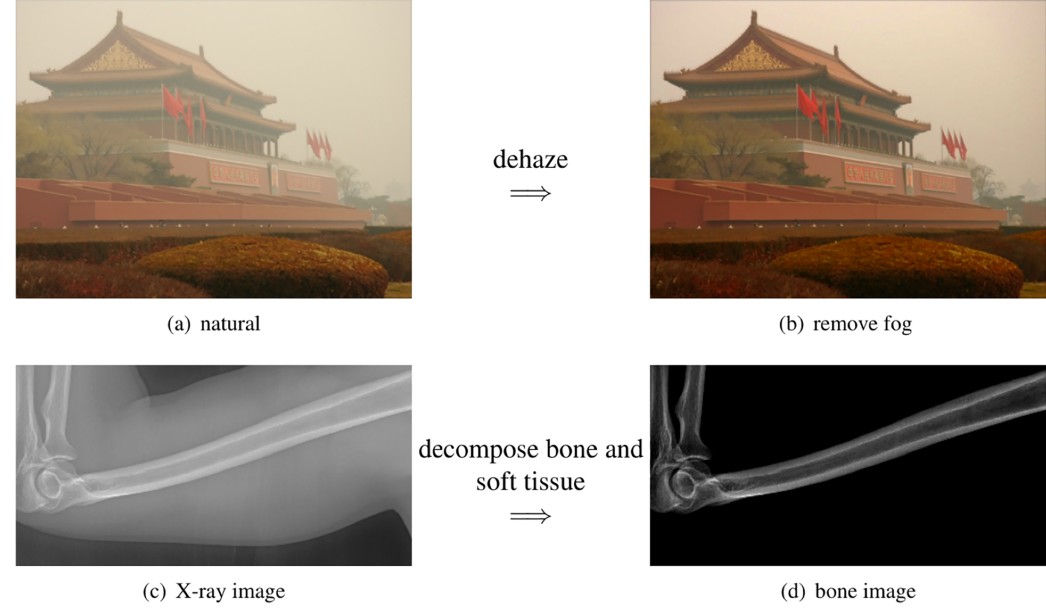

**Figure 2 The scattered light in X-ray images share the same physics as the natural images.** (A) The building is surrounded by the fog. (B) The fog can be removed by computation algorithms. (C) The bone is surrounded by the soft tissue. (D) The soft tissue can be removed by our method.

(*Van de Hulst, 1958*). Mie scattering, which involves spherical particles, was later studied by Mie and has been named after him. These fundamental principles of physics form the basis for modern image dehazing techniques.

## Scattered light in natural images

A typical example of scattering light in nature is the fog. As a result, the image from foggy natural scene is not clear. Image processing algorithms that virtually remove the fog are called dehazing (as shown in the top row of Fig. 2).

For natural images, the dehazing mathematical model is *Fattal (2008)*, *He, Sun & Tang (2011)*

$$f(x, y) = J(x, y)t(x, y) + A(1 - t(x, y)), \tag{1}$$

where $f$ is the observed image, $J$ is the unknown clear image to be estimated, $t$ is the transmission map to be estimated, and $A$ is the global atmospheric light to be estimated.

In recent years, there has been significant progress in improving the visibility of hazy images. These methods can be divided into three categories: simple contrast enhancement methods (*Tan, 2008*; *Fattal, 2008*), dark channel based methods (*He, Sun & Tang, 2011*) and deep learning methods (*Cai et al., 2016*).

Initially, the focus was on enhancing the contrast of foggy images, assuming that they lacked sufficient contrast (*Tan, 2008*; *Fattal, 2008*). However, these methods often require extensive computation and may result in noticeable artifacts.

A significant breakthrough came with the introduction of the dark channel prior in dehazing methods (*He, Sun & Tang, 2011*). The dark channel prior assumes the existence of a region with low-intensity values in the local neighborhood. This prior can be efficiently solved using the guided image filter, which has popularized the use of the dark channel prior in dehazing.

Deep learning is another approach for reducing scattering light in images (*Cai et al., 2016*; *Engin, Genc & Ekenel, 2018*). It assumes the availability of paired clear and foggy images, and a neural network can be trained to learn the mapping from foggy images to their corresponding clear images. However, in practical applications, acquiring paired images can often be challenging.

## Bone suppression or enhancement

Instead of interested by bones, some applications focus on the soft tissue such as pneumonia. For these applications, they try to reduce the visualization of bones. Such task is called bone suppression (*Suzuki et al., 2006*; *Chen & Suzuki, 2014*; *Li et al., 2020*).

In such bone suppression, previous approaches assume that the observed image $f$ is linearly composed by soft tissue $f_{\text{tissue}}$ and $f_{\text{bone}}$ as *Von Berg et al. (2016)*

$$f = f_{\text{tissue}} + f_{\text{bone}}. \tag{2}$$

Surely, such linear composition is fundamentally different from the nonlinear model in Eq. (1).

Moreover, such methods require strong geometric prior information about the imaging objects, such as the rib shapes for chest X-ray images. And they usually require to exactly find the bone boundaries (bone segmentation). Due to such geometric prior and accurate segmenation requirements, these methods are difficult to be extended from one imaging object to another. For example, the methods developed for ribs can not easily be used for feet or knees.

Even with the accurate bone segmentation, the resulting soft tissue images may have obvious artifacts because their linear composition assumption, Eq. (2), is not always valid. In contrast, the nonlinear model Eq. (1) has been shown effective in removing the haze in natural images.

These limitations motivate us to develop a new and generic mathematical model. Instead of suppressing bones or soft tissue, our model decompose one X-ray image into one soft tissue image and one bone image. These two images have exactly the same imaging domain. Therefore, our task is fundamentally different from bone enhancement task and bone suppression task.

## Motivation and contributions

The soft tissue in human body usually scatters the X-ray, severely reducing the quality of bone details in the resulting images. This fact motivates us to construct a novel mathematical model that can decompose bones and the soft tissue in X-ray images. The decomposed soft tissue image can be used for its related study such as pneumonia. The

bone image can be adopted for its related research such as bone fracture and bone age estimation.

The scattering light in X-ray images by the soft tissue shares the same physical law as the fog in natural images (*Gong & Sbalzarini, 2016*; *Gong, 2015*). Thus, the dehazing methods that have been developed for natural foggy images must be also valid on X-ray images (*Gong et al., 2019*). We transform the dehazing model into a new form for X-ray images with proper assumptions.

Different from the bone enhancement or suppression, we propose to decompose the input X-ray image into one bone image and one soft tissue image. Such task is named as bone and tissue decomposition (BTD). We construct a new mathematical model that can effectively decompose a single X-ray image into one soft tissues image and one bone image. Be aware the difference between our model and the bone segmentation task. Bone segmentation separates the imaging domain into bone region and background region (without overlap). However, our background and bone images share the same imaging domain (exactly overlapped with the same imaging domain). Such difference is illustrated in Fig. 3.

Our contributions are in the following folds:

- We propose a new image processing task named BTD.
- We propose a new mathematical model for BTD. This model is based on the well known image dehazing model, but with proper improvements for X-ray images.
- With proper assumptions, the BTD model leads to a standard Laplace equation, which can be efficiently solved.
- The resulting bone image is theoretically guaranteed to have larger image contrast than the original image.

Portions of this text were previously published as part of a preprint (https://arxiv.org/abs/2007.14510).

# BONE TISSUE DECOMPOSITION MODEL

In this section, we first show the novel mathematical model that decomposes the soft tissue and the bones. Then, we analyze its parameter. Finally, we show its connection with the previous dehazing model and other related models.

## Mathematical equation

In this article, we propose a new mathematical model for X-ray image decomposition. Our model is mathematically rooted in the scatted light physics and the dehazing model for natural images in Eq. (1). More specifically, we propose following model for X-ray images:

$$f(x,y) = \frac{1}{\alpha}B(x,y)(1 - T(x,y)) + T(x,y), \tag{3}$$

where $f(x,y) \in [0,1]$ is the observed image, $B$ is the unknown bone image, $T(x,y)$ is the soft tissue image, $\alpha \geq 0$ is a scalar parameter.

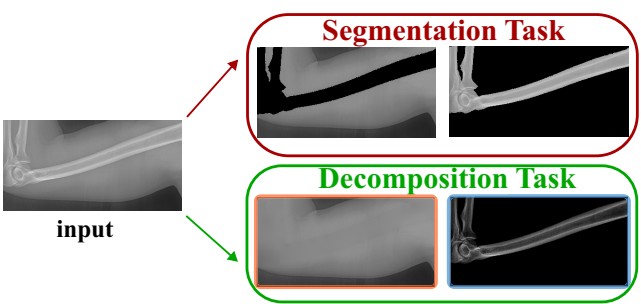

**Figure 3 Our task is different from the segmentation task.**

When $f(x, y) = T(x, y)$, it would force $B(x, y) = 0$. It means that the observation only comes from the soft tissue and there is no bones in the image. When $f(x, y) = \frac{1}{\alpha}B(x, y)$, it would force $T(x, y) = 0$, which indicates that the observation only comes from bones. Otherwise, the observation is composed by the tissue and bones images as the similar way in scattered light.

## Parameter $\alpha$ analysis

We use $\alpha$ as global constant variable, instead of spatially varying $\alpha(x, y)$. Although $\alpha(x, y)$ might achieve better visual result, it could introduce artifacts and it would lose the relationship between actual dose and image intensity in X-ray image. But when we use spatially constant $\alpha$, such linear scaling will keep such relationship between the actual physics and the intensity in X-ray images.

In later section, we will prove that $\alpha \geq 1$ (Eq. (11)), which theoretically guarantees to increase the image contrast. This property becomes clear when we set the background $T(x, y) = 0$. That is $f(x, y) = \frac{1}{\alpha}B(x, y)$. It means $\nabla B(x, y) = \alpha \nabla f(x, y)$, where $\nabla$ is the standard gradient operator. Since $\alpha \geq 1$, the contrast in bone image $B(x, y)$ is theoretically larger than the contrast in the input image $f(x, y)$. This theoretical property is numerically confirmed by all our experiments.

## Relationship with the dehazing model

Our model is different from Eq. (1) with two important changes. First, we define the soft tissue image (also called background image in this article) as

$$T(x, y) = A(1 - t(x, y)), \text{ where } A = 1. \tag{4}$$

Here, we assume $A = 1$. The reason is that only X-ray can reach the sensors (there is no other light resource). Second, we define the unknown bone image $B(x, y)$ as a linear scaling of $J(x, y)$

$$B(x, y) = \frac{1}{\alpha}J(x, y), \tag{5}$$

where $\alpha \geq 0$ is a scalar parameter. Such linear scaling keeps the physical meaning of the image intensity in X-ray images.

## Comparison with the linear model

Our model is nonlinear and inspired by scattered light models in physics and dehazing models for natural images. Once input X-ray image and its bone image are given, the soft tissue image can be computed as

$$T(x, y) = \frac{f - \frac{B}{\alpha}}{1 - \frac{B}{\alpha}}, \tag{6}$$

which is fundamentally different from the linear model, Eq. (2), in bone suppression (*Von Berg et al., 2016*)

$$f_{\text{tissue}} = f - f_{\text{bone}}. \tag{7}$$

To quantitatively demonstrate this differential performance, we implement both the nonlinear formulation (Eq. (6)) and its linear counterpart (Eq. (7)) using identical input radiographs $f$ and standardized bone images ($f_{\text{bone}} = \frac{B}{\alpha}$). The experimental validation employs a publicly accessible benchmark dataset (https://www.kaggle.com/raddar/digitally-reconstructed-radiographs-drr-bones) comprising 193 paired X-ray projections and their corresponding osseous segmentation maps. Subsequent soft tissue reconstructions derived from Eqs. (6) and (7) exhibit marked qualitative differences, as exemplified in Fig. 4. Critical evaluation of the resultant images, as annotated by directional indicators (red/green arrows), reveals that the nonlinear paradigm achieves significantly superior bone suppression efficacy relative to the linear approximation model.

Since the linear model and our model take the same input X-ray image and the same bone image, the difference in the estimated soft tissue images can only come from the models themselves. Therefore, the results in Fig. 4 numerically confirm that our model is better than the linear model Eq. (2).

## OUR NUMERICAL SOLVER

Now, our task is to numerically solve this model. In our nonlinear model, there are three unknown variables, $\alpha$, $B(x, y)$ and $T(x, y)$. We notice that the bone image $B(x, y)$ can be easily computed if the background image $T(x, y)$ is known. Therefore, we can solve our model by first finding the $T(x, y)$ and $\alpha$. We introduce some proper assumptions to solve our model. With these assumptions, our model leads to a standard Laplace equation, which has an efficient numerical solver.

## Assumptions

Since our model, Eq. (3), is ill-posed, we have to make some assumptions to solve this model. We make the following four assumptions to simplify the solving process.

- First of all, we assume $T(x, y) \leq f(x, y)$. This assumption makes sure that $U(x, y) \geq 0$.
- Second, we assume $0 \leq T(x, y) < 1$, which avoids the denominator to be zero. As a result, $\frac{1}{1-T(x,y)} > 1$, which helps in improving the bone image contrast as shown in later sections (Eq. (11)).
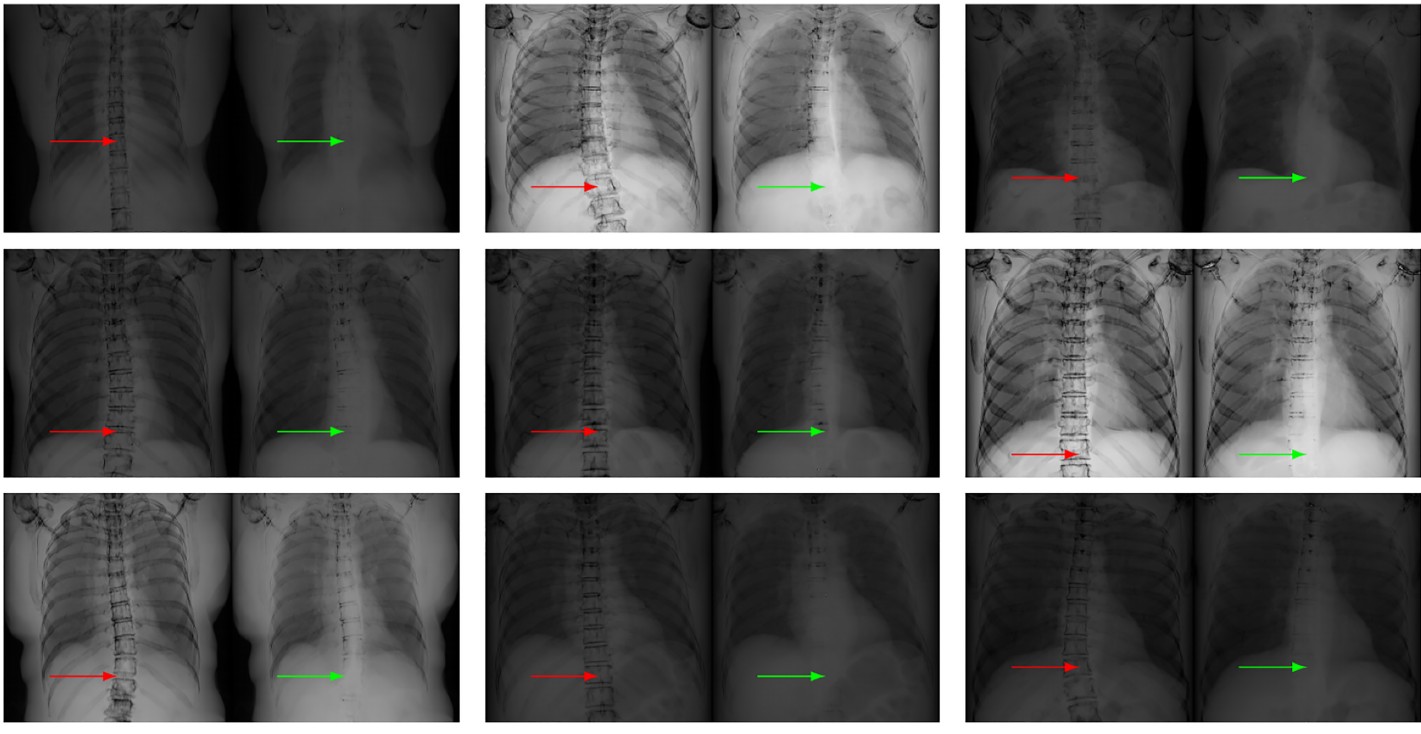

**Figure 4** **Our nonlinear model Eq. (6) (right in each panel) leads to less visible bones than the linear model Eq. (2) (left in each panel) on chest X-ray images, as indicated by the arrows.** In each case, the soft tissue image is computed from the given input X-ray image and its related bone image. Our model keeps less bones than the linear model, especially at the region indicated by the red and green arrows. This fact indicates that our model is better than the linear model in terms of bone suppression.

- Third, we assume that $T(x, y)$ is second order differentiable (*Gong & Goksel, 2019*; *Gong & Sbalzarini, 2017*). Such smoothness assumption is reasonable because the physical configuration of soft tissue is smooth.

- Fourth, we assume that the maximum value in $B(x, y)$ is one. Such assumption is used to determine the value of $\alpha$. In later section, based on this assumption, we can prove that $\alpha \geq 1$. Such theoretical result guarantees the bone image contrast enhancement as shown in later section.

## A flexible bone mask

Now, we have enough assumptions to find $T(x, y)$. We estimate the $T(x, y)$ by a two-step strategy. First, we roughly estimate a mask $M(x, y)$ that covers bones. Be aware that the mask $M(x, y)$ only needs to cover the bones. It does not necessarily be exactly aligned with bone boundaries as the image segmentation task.

Therefore, there are several ways to obtain such mask. First, it can be easily obtained by a simple threshold method followed by morphology operations. Second, it can also be estimated by active contour methods. Third, it can even be given interactively by users' input. In short, the way of obtaining this mask is flexible.

Moreover, the mask $M(x, y)$ itself is also flexible. We use three different masks for the same input image, as shown in each row of Fig. 5. Although the masks are varying, the resulting soft tissue images and bone images are similar. Be aware that our mask only needs to cover the bones. Its boundary is not necessarily aligned with the bone boundary (the image segmentation task).

## Soft tissue image

With a bone mask $M(x, y)$, we can find the soft tissue image $T(x, y)$ by solving a Laplace equation. Let $M(x, y)$ denote our mask. Now, we need to estimate the soft tissue intensity in this mask. This problem can be modeled as following minimization task

$$\min \int \int_M ||\nabla T||^2 \mathrm{d}x\mathrm{d}y \,, \text{s.t. } T_{\partial M} = f_{\partial M}, \tag{8}$$

where $\partial$ denotes the boundary. The optimal solution of this energy is the standard Laplace equation

$$\Delta T_M = 0 \,, \text{s.t. } T_{\partial M} = f_{\partial M}. \tag{9}$$

There are several efficient Poisson solvers available for this equation. We summarize these solvers in Table 1, where each algorithm's computational complexity is also shown.

We use the convolution pyramid method (*Farbman, Fattal & Lischinski, 2011*) to solve the this equation for two reasons. First, it is a direct method and so we do not have to iterate. Second, the pyramid method has linear computational complexity.

Thanks to the efficiency of Pyramid method, the solution of Eq. (9) can be easily obtained. The estimated soft tissue image is shown in Fig. 5C. The running time is 0.1 s in MATLAB on a ThinkPad P1 laptop with Intel Xeon E2176 CPU with 2.70 GHz. The image resolution is $1,022 \times 757$. Such performance (7.7 Mpixel/second) is fast enough for clinical applications in practice.

## Bone image

After estimating $T(x, y)$, we need to estimate $\alpha$ for bone image $B(x, y)$ estimation. As mentioned, we assume the maximum value in $B(x, y)$ is one. Therefore, we define

$$\alpha \equiv \frac{1}{\max\limits_{(x,y)} \left\{ \frac{f(x,y) - T(x,y)}{1 - T(x,y)} \right\}}. \tag{10}$$

Since $0 \leq f(x, y) \leq 1$, we have $\frac{f(x,y) - T(x,y)}{1 - T(x,y)} \leq 1$. As a result, we can prove

$$\alpha \geq 1. \tag{11}$$

This parameter linearly increases the contrast in the bone image. As mentioned, such linearity can keep the physical meaning of intensity in X-ray images.

Finally, the bone image can be computed as (shown in Fig. 5D)

$$B(x, y) = \alpha \frac{f(x, y) - T(x, y)}{1 - T(x, y)}. \tag{12}$$
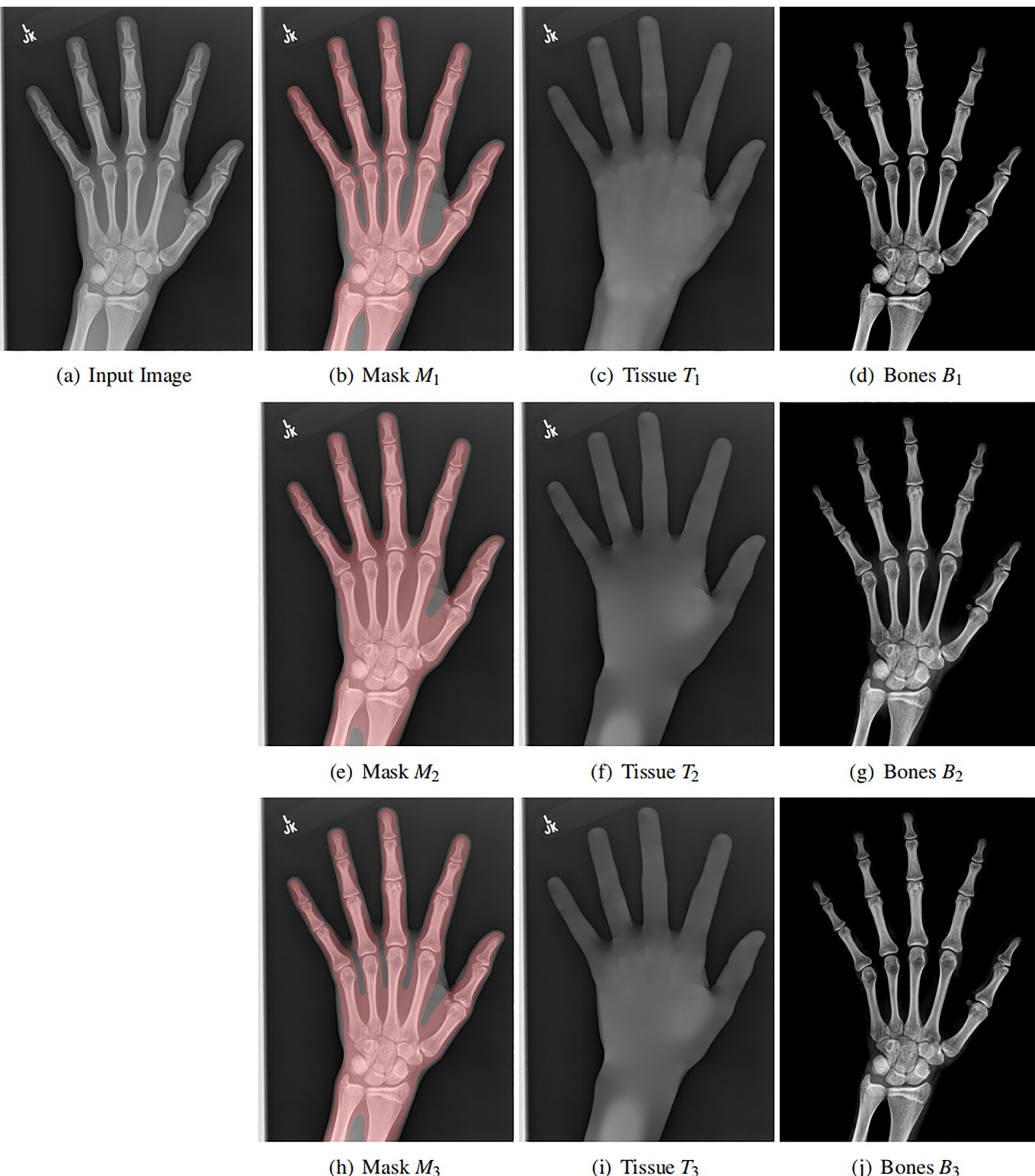

**Figure 5 Original X-ray image (A), our mask (B, E ,H), estimated background (C, F, I) and estimated bones (D, G, J).** The soft tissue image $T(x, y)$ is obtained by solving a Laplace equation from the mask. Each row contains different mask, showing the flexibility of the mask. In all cases, the bone image $B(x, y)$ has better contrast than the input image.

**Table 1  Summary of Laplace solvers for $N$ samples.**

| Solver | Cholesky | Jacobi | GaussSeidel | SOR |
|---|---|---|---|---|
| Type | Direct | Iterative | Iterative | Iterative |
| | $O(N^3)$ | $O(N^2)$ | $O(N^2)$ | $O(N^{3/2})$ |
| Solver | FFT | Multigrid | Wavelet | Pyramid |
| Type | Direct | Iterative | Direct | Direct |
| | $O(N \log N)$ | $O(N)$ | $O(N)$ | $O(N)$ |

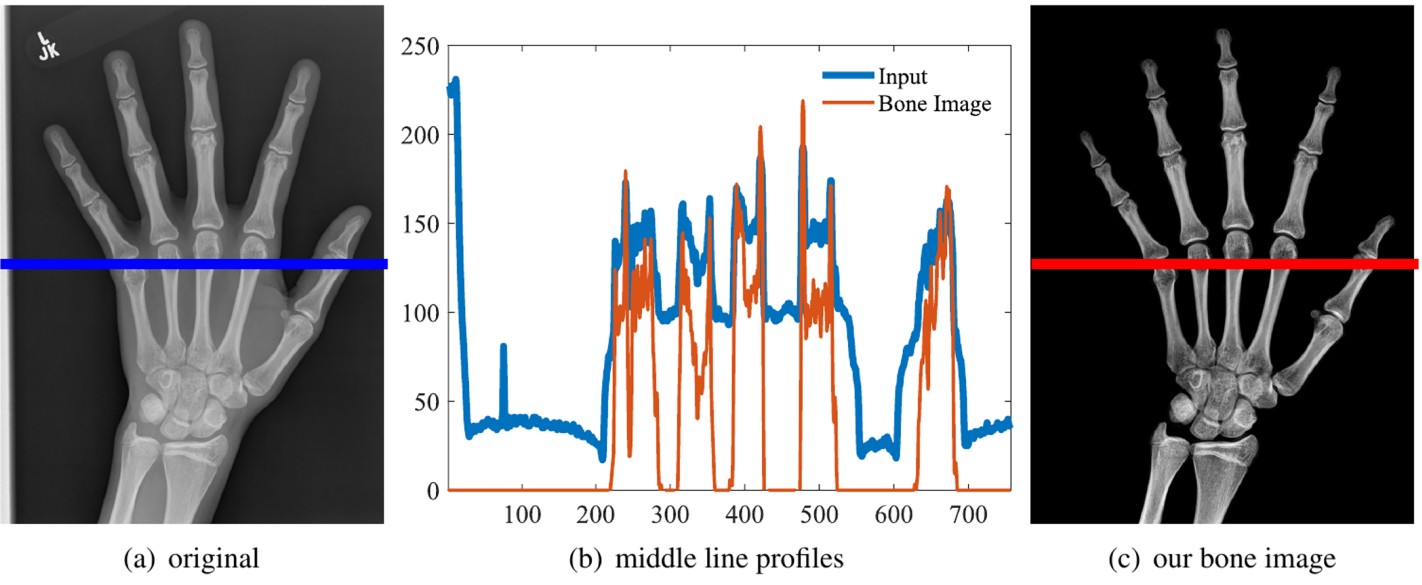

(a) original         (b) middle line profiles         (c) our bone image

**Figure 6  Original image (A), the line intensity profile from the input (blue) and our result (red) (B), our estimated bone image (C).** In this case, $\alpha = 1.44$ and the contrast (gradient) is increased, although the intensity might be lower.     

One example is shown in the first row of Fig. 5. And the middle line intensity profiles of original and our results are shown in Fig. 6. In this case, $\alpha = 1.44$ and the image contrast is enhanced. As shown in Eq. (11), $\alpha \geq 1$ and the enhancement is theoretically guaranteed. All our experiments also confirm this property.

**Guaranteed contrast enhancement**

With our model, we can prove the following theorem

**Theorem 1.** *The bone image $B(x, y)$ has larger image contrast than the input image $f(x, y)$ for most pixels.*

*Proof.* To show the image contrast enhancement, we need to study the gradient of the resulting bone image. From Eq. (12), we can get the gradient of $B$

$$\nabla B = \alpha \left[ \frac{\nabla f}{1 - T} - \frac{1 - f}{(1 - T)^2} \nabla T \right]. \tag{13}$$

---

**Algorithm 1  Bone and tissue decomposition.**

**Require:** input X-ray image *f(x,y)*
   (1) obtain the mask *M(x,y)* by active contour or user input
   (2) compute *T(x,y)* by solving Eq. (9)
   (3) compute α by Eq. (10)
   (4) compute *B(x,y)* by Eq. (12)
**Ensure:** *T(x,y)*, *B(x,y)*

---

Since we assume $T(x, y)$ is smooth, we know that $\nabla T(x, y) \approx 0$ for most locations (*Gong & Sbalzarini, 2016*) (also see the gradient statistics in Fig. 5 from *Gong & Goksel (2019)*). Therefore, we have

$$\nabla B \approx \alpha \frac{\nabla f}{1 - T} \geq \alpha \nabla f \geq \nabla f. \qquad (14)$$

The last inequality comes from the $\alpha \geq 1$ in Eq. (11). This result indicates that the bone image has better image contrast than the input image for most of pixels.

Such theoretical guarantee is important for practical applications, where the robustness of the method is concerned. Our method makes sure that the bone image is clearer than the original input. Moreover, the larger $\alpha$, the better bone image contrast. The $\alpha$ depends on the X-ray source and the imaging objects.

### The complete algorithm

In summary, our model in Eq. (3) can be efficiently solved by Algorithm 1.

## EXPERIMENTS

We performed four experiments for our method. First, we perform our method on several X-ray images that contain different types of bones, showing our method is not restricted by specific imaging objects. Second, we compared our method with image enhancement method and dehazing method, showing that our model works better than simple image enhancement and dehazing model. Third, we compared our method with bone suppression and enhancement methods, showing that our method works better in both cases. Fourth, we perform our method on a hand X-ray image dataset, showing its effectiveness and efficiency on high resolution images in practical applications.

### Different imaging objects

Our model is not restricted by any imaging object. It can work on various bone X-ray images. Several results from our method are shown in Fig. 7, including knees, arm, hand, *etc*. The left column is the original input image. The right two columns are the resulting soft tissue and bone image from our method, respectively. It can be told that the soft tissue image is smooth as we assumed. Meanwhile, the bone image has better image contrast as desired.

Moreover, our method can reach real-time performance on these X-ray images. The running time of our method on these images is reported in Table 2. Our method is fast

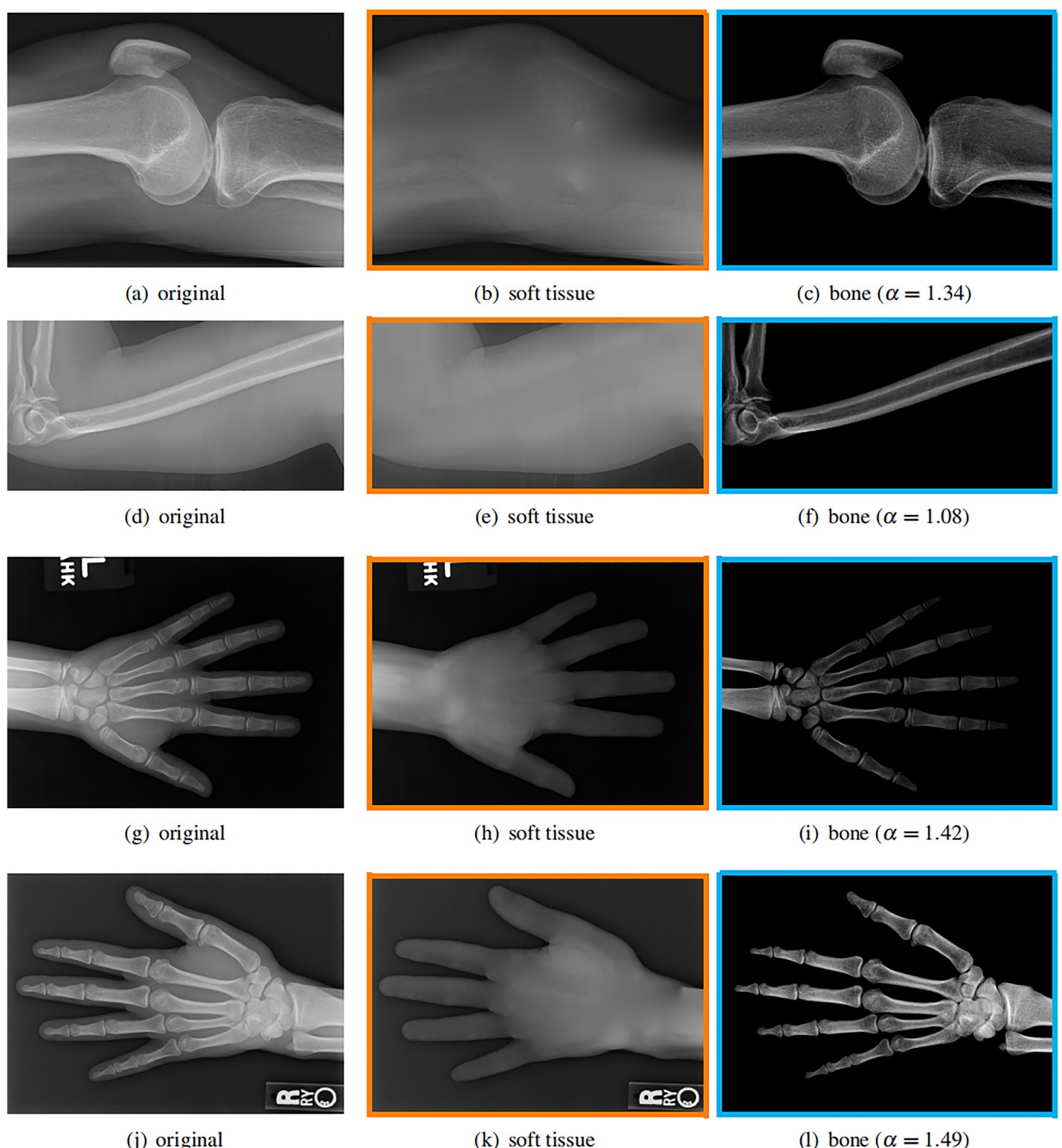

(a) original     (b) soft tissue     (c) bone ($\alpha = 1.34$)

(d) original     (e) soft tissue     (f) bone ($\alpha = 1.08$)

(g) original     (h) soft tissue     (i) bone ($\alpha = 1.42$)

(j) original     (k) soft tissue     (l) bone ($\alpha = 1.49$)

**Figure 7 Our method works well on different imaging objects.** Input X-ray images (A, D, G, J), our estimated soft tissue (B, E, H, K) and estimated bone image (C, F, I, L).

| Image | α | Resolution | Time (seconds) | Performance |
|---|---|---|---|---|
| **Table 2 The running time in seconds of our algorithm.** | | | | |
| Figure 7A | 1.34 | 319 × 442 | 0.031 | 4.5 MP/s |
| Figure 7D | 1.08 | 193 × 382 | 0.019 | 3.9 MP/s |
| Figure 7G | 1.42 | 514 × 711 | 0.094 | 3.9 MP/s |
| Figure 7J | 1.49 | 336 × 471 | 0.041 | 3.9 MP/s |

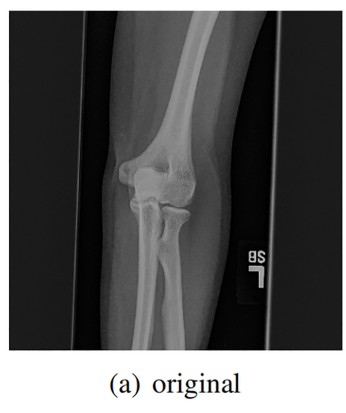 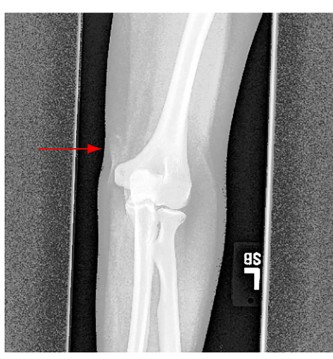 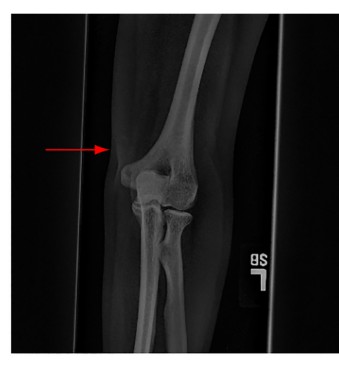 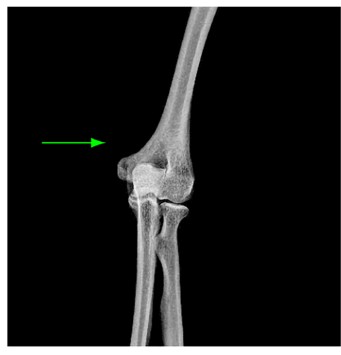

(a) original  (b) HistEqGonzalez and Woods (2006)  (c) dehazing He et al. (2011)  (d) ours α = 2.08

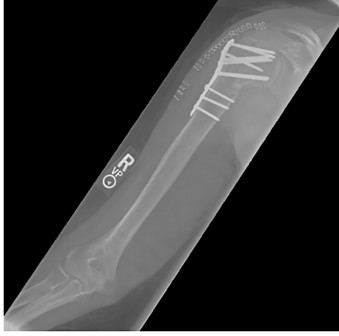 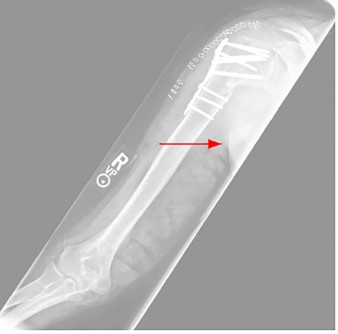 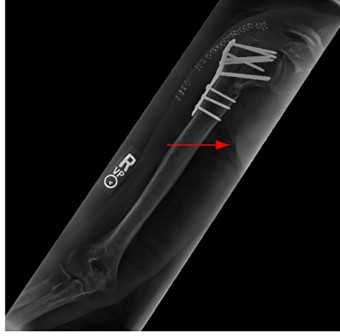 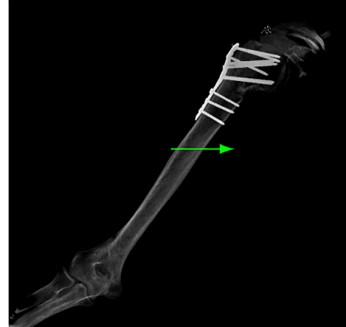

(e) original  (f) HistEqGonzalez and Woods (2006)  (g) dehazing He et al. (2011)  (h) ours α = 1.43

**Figure 8 Compare our method with other enhancement methods.** Left to right: original images, image enhancement by histogram equalization (*Gonzalez & Woods, 2006*), results from dehazing method with dark channel prior (*He, Sun & Tang, 2011*), and results from our method. The conventional methods can not completely remove the soft tissue (red arrows). Our method removes the soft tissue (green arrows) and has better contrast.     

enough for these low resolution images. For high resolution images, the running time will be shown in later section. The experiments are performed in MATLAB 2022b (The MathWorks, Natick, MA, USA) on a laptop with Intel Xeon E2176 CPU with 2.70 GHz.

## Comparison with dehazing and enhancement

We further compare our method with a classical image enhancement method and a dehazing method for natural images (*He, Sun & Tang, 2011*), which uses dark channel prior. We tested on ten images and two of them are shown in Fig. 8. The classical image

![PeerJ]

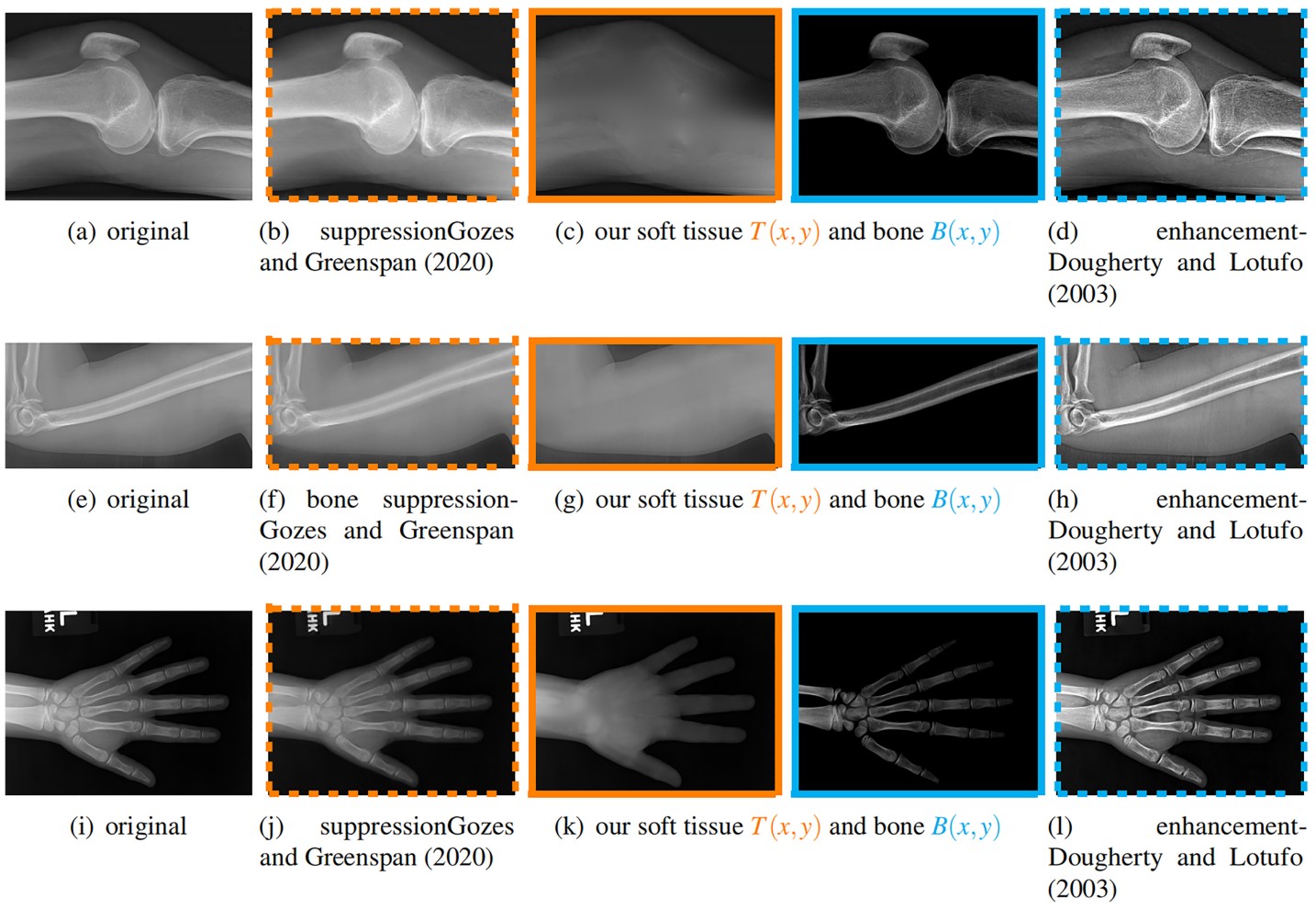

(a) original    (b) suppressionGozes and Greenspan (2020)    (c) our soft tissue $T(x, y)$ and bone $B(x, y)$    (d) enhancement-Dougherty and Lotufo (2003)

(e) original    (f) bone suppression-Gozes and Greenspan (2020)    (g) our soft tissue $T(x, y)$ and bone $B(x, y)$    (h) enhancement-Dougherty and Lotufo (2003)

(i) original    (j) suppressionGozes and Greenspan (2020)    (k) our soft tissue $T(x, y)$ and bone $B(x, y)$    (l) enhancement-Dougherty and Lotufo (2003)

**Figure 9** **From left to right: original, bone suppression (*Gozes & Greenspan, 2020*), our soft tissue and bone image, bone enhancement (*Dougherty & Lotufo, 2003*).** The images with the same color frame are comparable. These results confirm that our method works better for both tasks.

enhancement method (histogram equalization) enhances both the soft tissue and bones. Such nonlinear process losses the relationship between image intensity and physical X-ray dose.

Our model is also different from the dehazing model. The dehazing method for natural images can not completely remove the soft tissue in X-ray image, as shown by the red arrows in Fig. 8. In contrast, our method does not have this issue. This is because we estimate a better soft tissue image. Moreover, our bone image has better image contrast, which is theoretically guaranteed as described.

## Comparison with bone suppression and bone enhancement

We compare our soft image with a bone suppression method (*Gozes & Greenspan, 2020*) and compare our bone image with a bone enhancement method (*Dougherty & Lotufo, 2003*). We tested them on ten images and three results are shown in Fig. 9, where the left column is the original input image.

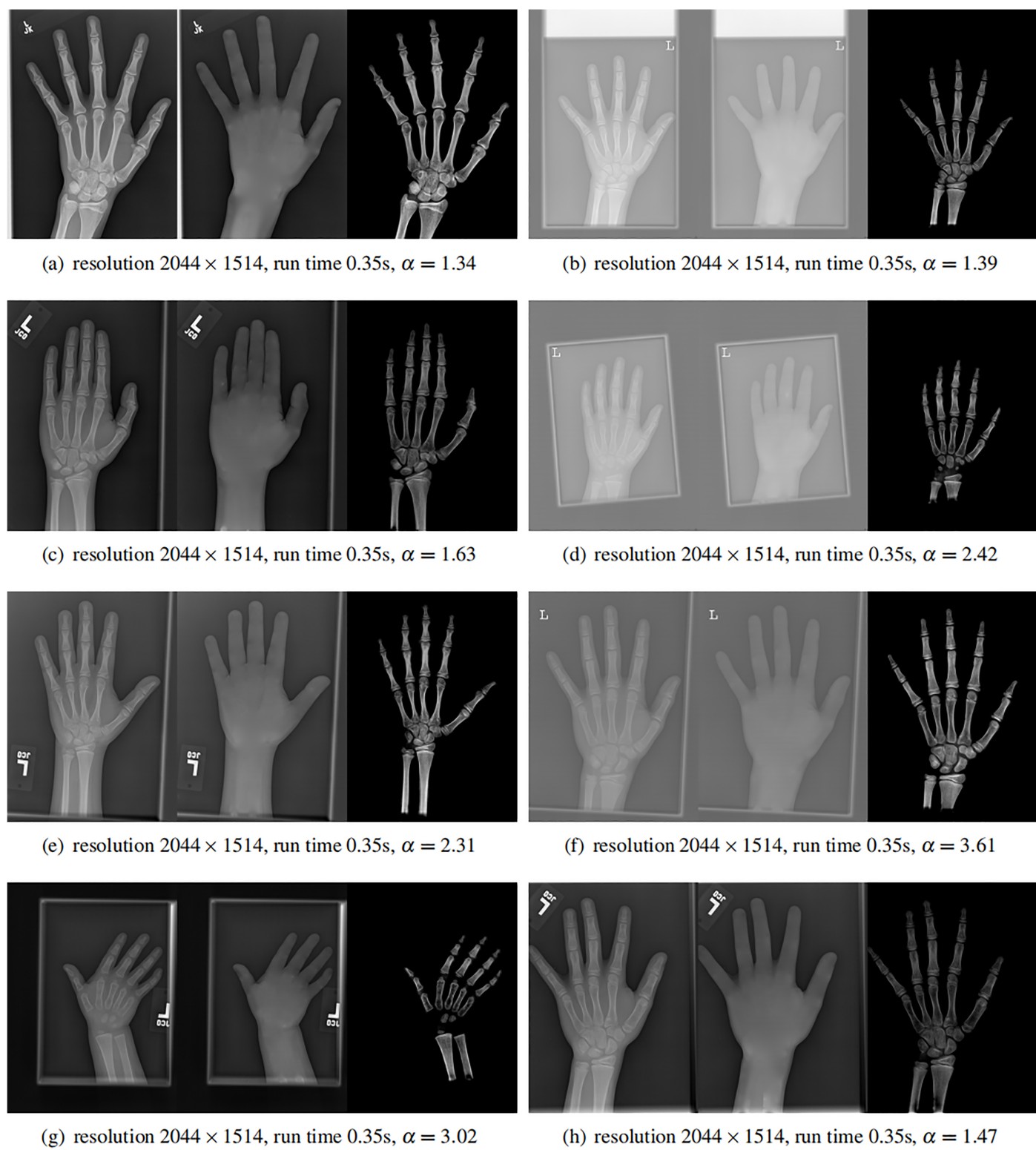

(a) resolution 2044 × 1514, run time 0.35s, $\alpha = 1.34$

(b) resolution 2044 × 1514, run time 0.35s, $\alpha = 1.39$

(c) resolution 2044 × 1514, run time 0.35s, $\alpha = 1.63$

(d) resolution 2044 × 1514, run time 0.35s, $\alpha = 2.42$

(e) resolution 2044 × 1514, run time 0.35s, $\alpha = 2.31$

(f) resolution 2044 × 1514, run time 0.35s, $\alpha = 3.61$

(g) resolution 2044 × 1514, run time 0.35s, $\alpha = 3.02$

(h) resolution 2044 × 1514, run time 0.35s, $\alpha = 1.47$

**Figure 10 From left to right in each panel: input X-ray images (left), our estimated soft tissue (middle) and estimated bone image (right).** The resolution, running time of our algorithm and parameter $\alpha$ are provided. For these practical images, our method requires about half second to achieve the bone and soft tissue decomposition task in MATLAB language on a laptop with Intel Xeon E2176 CPU with 2.70 GHz.

For the soft tissue image, our method obtains a result which does not have obvious bones on various X-ray images, including knees, humerus, and hands. In contrast, the method from *Gozes & Greenspan (2020)* does not suppress the bone much.

For the bone image, our method gives an image that only contains bones. In contrast, the bone enhancement method (*Dougherty & Lotufo, 2003*) enhances the bones but still keeps some soft tissue in the result.

### High resolution images

Finally, we applied our method on a hand X-ray image data set (RSNA), which contains more than 10,000 hand X-ray images. The image has high resolution (usually larger than $1,514 \times 2,044$). These images are collected from clinical applications. Therefore, we can test the performance of our method on practical images, showing its efficiency and effectiveness.

In each panel of Fig. 10, the input image (left) is decomposed into soft tissue (middle) and bone image (right) by our method. Although we only show several images from the data set, the results for the rest images are similar.

The bone images have better image contrast since the parameter $\alpha \geq 1$ is theoretically guaranteed. Since details on bones become clear, such enhancement can benefit the bone diagnosis in practice.

Moreover, the running time of our method on such high resolution images is less than half second in the MATLAB language on a laptop. Therefore, it can be easily deployed in real applications. If higher performance is required, our model can be solved by the parallel Laplace equation solver on a modern graphic process unit (GPU), which usually has thousands of cores.

We believe that such bone and soft tissue decomposition model is important for X-ray images, bone study, soft tissue diagnosis, *etc*. Despite the nice mathematical properties of the model, it can be very efficiently solved by solving a standard Laplace equation.

## CONCLUSION

In this article, we propose to decompose one X-ray image into a soft tissue image and a bone image. We name this task bone and tissue decomposition, for which a novel mathematical model is developed. Our mathematical model is inspired by the natural dehazing model, but with proper extension for X-ray images.

With several assumptions, our model leads to a Laplace equation, which can be efficiently solved. Solving the 2D Laplace equation is a classical problem. And we use the wavelet solver developed in *Farbman, Fattal & Lischinski (2011)* to solve this equation. After solving this equation, we obtain the soft tissue image. With the soft tissue image and the original input image, we can compute the bone image with a close form solution expression. The bone image is uniquely determined by the soft tissue image.

The resulting bone images are theoretically guaranteed to have better image contrast (larger gradient) because of $\alpha \geq 1$. Several numerical experiments have confirmed this

property. Better image contrast is important for clinical diagnosis, such as bone fracture and surgery planning.

Our method can enhance the details on bones in X-ray images, without losing the relationship between the intensity and actual physical X-ray received on the sensor. This property is different from the conventional image enhancement methods. Our result can improve other bone related tasks, such as bone segmentation, recognition, diagnosis, surgery planning, *etc*.

Moreover, our method is numerically fast. It can process 8.8 million pixels per second in MATLAB software on a ThinkPad P1 laptop with Intel Xeon E2176 CPU. For real X-ray images with resolution $2,044 \times 1,514$, our method only requires 0.35 s to finish the bone and soft tissue decomposition task. In practice, this performance can be further improved by C++ language on a better hardware. If higher performance is required, our model can be solved by parallel algorithms on modern graphic processing unit (GPU), which usually contains thousands of cores.

Our method can be applied in a large range of applications. It can be used for bone study, for example, bone fracture diagnosis. It can also be used in bone age assessment, reducing the influence of soft tissue. Our method can also be used for applications where the soft tissue is the main concern, for example, pneumonia in chest X-ray images. Our method can be used as a pre-processing approach for deep learning training data set preparation.

Our method assumes the homogeneity inside the mask, which is not always valid. When the mask contains complex geometries, our method might not generate the accurate soft tissue image, and thus lead to artifacts in the bone image. This issue can be tackled by recent deep learning methods.

In the future, we plan to solve our mathematical model by modern convolution neural networks (CNN). Thanks to their excellent achievements in the past few years, CNN have been used in many different image processing and computer vision tasks. There are several advantages to use CNN in this task. First, CNN can handle complex objects without knowing the mask. Second, the CNN can be robust with the noise. Third, the CNN can be trained to be adaptive to the input dataset. In the CNN, the loss function can be constructed from our mathematical model proposed in this article. And the training ground truth for the CNN can be our results from X-ray images. More specifically, the network can be trained on the paired data $(f_i, T_i, B_i)$, where $f_i$ is the input image and $S_i$, $U_i$ are results from our method. Thanks to the power of neural networks, the CNN will achieve higher accuracy in generating the soft tissue and bone images from a single input X-ray image (*Gong, Paul & Sbalzarini, 2012*; *Yu & Orchard, 2019*; *Gong & Sbalzarini, 2013*; *Yin, Gong & Qiu, 2019a*; *Gong, 2015*; *Yu et al., 2022*; *Gong & Sbalzarini, 2017*; *Zong et al., 2021*; *Gong et al., 2018a, 2018b*; *Gong & Goksel, 2019*; *Yin, Gong & Qiu, 2019b*; *Gong & Chen, 2019*; *Gong et al., 2019*; *Gong, 2019, 2022*; *Yin, Gong & Qiu, 2020*; *Gong & Chen, 2020*; *Gong et al., 2021b*; *Tang, Gong & Qiu, 2023*; *Gong, 2024*; *Gong et al., 2021a*; *Gong & Lin, 2024*; *Wei, Tang & Gong, 2024*; *Gong, 2025a, 2025b*).

### Funding

This research was funded by National Natural Science Foundation of China (grant number 12471502 and 61907031), Shenzhen Science and Technology Program (grant number 20231121165649002, JCYJ20220818100005011 and 20231127143250002). The funders had no role in study design, data collection and analysis, decision to publish, or preparation of the manuscript.

### Grant Disclosures

The following grant information was disclosed by the authors:
National Natural Science Foundation of China: 12471502, 61907031.
Shenzhen Science and Technology Program: 20231121165649002, JCYJ20220818100005011, 20231127143250002.

### Competing Interests

The authors declare that they have no competing interests.

### Author Contributions

- Zhili Wei conceived and designed the experiments, performed the experiments, analyzed the data, authored or reviewed drafts of the article, and approved the final draft.
- Wenming Tang performed the experiments, prepared figures and/or tables, authored or reviewed drafts of the article, and approved the final draft.
- Yuanhao Gong conceived and designed the experiments, analyzed the data, prepared figures and/or tables, authored or reviewed drafts of the article, and approved the final draft.

### Data Availability

The Matlab code for the algorithm is available in the Supplemental File.

### Supplemental Information

Supplemental information for this article can be found online at http://dx.doi.org/10.7717/peerj.20016#supplemental-information.

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
