# Peer review of "Bone-tissue decomposition of a single X-ray image via solving a Laplace equation"

_PeerJ, doi:10.7717/peerj.20016_

## Round 0.1 · original submission · Major Revisions

Substantial revisions are necessary before we can reconsider your manuscript.

Reviewer 1 ·

Basic reporting

The Authors submitted a paper regarding a novel approach in bone tissue decomposition for X-ray images. The separation of bone and soft tissue components is useful for several clinical applications, and for this reason, many papers have been published about it. This paper describes a new model comparing the performance with the algorithms already used for this purpose.
The article is written in a clear and understandable way.
The English language needs a few revisions.

However, there are some criticisms reported in the following sections.

Line 28: the references used “Huang et al. (2023); Ataei et al. (2024); Gong et al. (2019).” Don’t meet the sentence “X-ray has been widely used in the field of biomedical imaging and clinical diagnosis”, but they are all focused on other issues that just slightly cover X-ray usage in imaging and diagnosis. Please review all the references in the paper for correct usage.

The introduction should be better referenced. (e.g, reference should be used for sentences at lines 48-50, 59-62, 63-65,66-68…).

Line 47: “challenge for medical professionals and researchers to see them clearly or study them in detail”…the main focus should be the clinical aspect, so it should be avoided to use researchers to justify the use of this new approach.

Line 48-54: This section should be improved with more clinical detail regarding X-ray importance and applications (not only related to bone investigations).

Line 70: “This complexity presents challenges for clinical diagnosis.” Please give some more details and explanations regarding this statement.

Figure 4: “This fact indicates that our model is better than the linear model in terms of bone suppression.” Did you involve a radiologist for data interpretation and better understand the clinical value of the resulting images? Even though bone vertebrae in the images are less evident, it should be used as a clinical evaluation to state that the model is working better in bone suppression, because the model could cause some other artifacts. Please rephrase.

Line 192-195: Please rephrase considering:
- The use of a clinician to evaluate the resulting images in the different approach
- The criteria used to state which model is better
- The criteria were not considered to make the comparison between the resulting images

The code is complete and working accordingly to the results presented in the paper.

Experimental design

In the experiments section it should be stated the origin of the datasets used with a proper link if necessary and if publicly available.

Line 275-276: It should be listed exactly how many images for each type have been analyzed. Please avoid etc. listing the various anatomic sites.

Line 282: Please provide the MATLAB version used.

Line 285-286: it should be listed exactly how many images for each type have been analyzed, comparing the methods. Have you ever noticed an artifact in your method? Do you have images of other anatomical sites where internal organs could be visible in an X-ray image (e.g., lumbar or sacral vertebrae)?

Line 294: It should list exactly how many images for each type have been analyzed, comparing the methods.

Validity of the findings

Line 169-170: “the relationship between actual dose and image intensity”:
It is not clear if the dose is the only component that contributes to local image intensity. The scattering component of X-ray, and disomogeneity of the detector/beam or whatever is part of the original image, appear not to be corrected, and in any case seem to be part of the decomposed images of bone and soft tissue.
Notwithstanding this, at lines 129-131, you stated that x-ray scatter is what motivates the construction of this model..
Several mathematical models have tried to remove scatter from beam geometry, object density, and others based on Monte Carlo simulations, but your method seems to bypass this consideration.
Please clarify.

Line 275: “Our model is not restricted by any imaging object. It can work on various bone X-ray images.”.
Did you try to run the model on both “RAW” and “FOR PRESENTATION” images?
How can geometry and energy of the beam affect the results?
These aspects should be investigated deeper.

Line 334-335: “without losing the relationship between the intensity and actual physical X-ray received on the sensor”. In the paper, the performance of the method is evaluated by visual evaluation of bone and tissue, so if another model that doesn’t preserve this relation, but performs better in bone/tissue decomposition, would be better?
It is not clear what the added value is of still having a relationship between the intensity and the actual physical X-ray received on the sensor. Please explain better why this is important.

Line 349-353: It is not clear what the advantage (clinical and technical) is of using CNN based on the reported approach.

Additional comments

The Authors submitted a paper regarding a new approach in bone tissue decomposition for X-ray images. The separation of bone and soft tissue components is a very promising area, and there are a lot of clinical applications. Depending on the focus of the clinical application (bone or soft tissue), the criteria to evaluate the performance of a model could vary, and therefore, it is not easily definable in the initial stages of validation.

The method presented by the authors is interesting, but needs some integrations and corrections.

Reviewer 2 ·

Basic reporting

-

Experimental design

-

Validity of the findings

-

Additional comments

1. Within the scope of the study, a method that is stated to be computationally fast and has high contrast has been proposed for bone-tissue decomposition in X-ray images.

2. In the introduction, the importance of the subject, scattered light, bone enhancement and suppression, and x-ray images are mentioned at a basic level. In this section, the literature review section should be made a little more in-depth, and the main contributions section stated in the items should be detailed in a way that emphasizes and highlights its originality.

3. When the mathematical equation, comparison with the linear model and relationship with the dehazing model specified regarding the proposed bone-tissue decomposition model are examined in detail, it is observed that it has a certain level of originality.

4. More detailed information should be provided regarding the dataset of bone x-ray images used in the study. In terms of the dataset type, experiments on more bone types will increase the quality of the study.

5. For the study to prove itself in terms of applicability, it is recommended to conduct more experiments. In addition, there are serious deficiencies in terms of the analysis of the results and metric analysis.

6. Although it is stated in the Future works section that the proposed model can be applied to convolutional neural networks, a more detailed explanation should be made in terms of usability, and its superiority over existing loss functions should be stated more clearly.

As a result, the study can make an important contribution to the literature on bone-tissue decomposition, but attention should be paid to the above items.

Reviewer 3 ·

Basic reporting

Through the construction of mathematical calculation formulas, using a post-processing program or project, the authors tried to decompose an X-ray image into two pictures showing only bones and only soft tissues. The authors believed that obtaining an image only showing skeletal structures with higher resolution would be more conducive to clinical application and diagnosis.

The ability to decompose an X-ray image clearly showing the structure of the human body (bone and soft tissue) by selecting an appropriate calculation formula and using computer post-processing may seem cool, but from the perspective of a clinician, it has no value for disease diagnosis and evaluation. The structure of the article is not reasonable, and the theoretical basis and the basis of formula deduction are not explained.

Experimental design

-

Validity of the findings

Line 63-65 “In clinical applications, the Window Technique is commonly used to limit the intensity of bone within a specific range. This effectively removes soft tissue regions that do not overlap with bones. However, this method cannot remove soft tissue that overlaps with bone regions.” Can you give an explanation about why we need to remove soft tissue that overlaps with bone regions? For ease of diagnosis, or good looks?

Line 193- 195 “Since the linear model and our model take the same input X-ray image and the same bone image, the difference in the estimated soft tissue images can only come from the models themselves. Therefore, the results in Fig. 4 numerically confirm that our model is better than the linear model Eq. (2).” Why does Figure 4 show that the right picture is better than the left one? Can you give objective indicators or some measurement values? Only with visual judgment is it too informal.

---

## Round 0.2 · Minor Revisions

Please review and make the necessary changes suggested by the first reviewer before we can accept your paper for publication.

Reviewer 1 ·

Basic reporting

(2) Line 47: “challenge for medical professionals and researchers to see them clearly or study them in detail”…the main focus should be the clinical aspect, so it should be avoided to use researchers to justify the use of this new approach.

Thank you for the feedback. We have now fixed this issue by changing the sentence to “challenge for medical professionals and researchers to see the details on bones”

As stated previously, please just use medical professionals and not researchers

(3) Line 48-54: This section should be improved with more clinical detail regarding X-ray importance and applications (not only related to bone investigations).

Thank you for the feedback. We have improved this section.

It doesn’t seem to be improved.

Experimental design

(4) Line 70: “This complexity presents challenges for clinical diagnosis.” Please give some more details and explanations regarding this statement.

Thank you for the feedback. We have changed the sentence to “Such a complex or unknown relationship causes artifacts and makes the results more difficult to interpret.”

Not enough, examples and references are needed for this statement

Validity of the findings

(6) Line 192-195: Please rephrase considering:
- The use of a clinician to evaluate the resulting images in the different approach
- The criteria used to state which model is better
- The criteria were not considered to make the comparison between the resulting images

Thank you for the feedback. We rewrite these paragraphs.

Cannot find where these paragraphs has been rewritten, please add limitations of your methods in the conclusions.

Reviewer 2 ·

Basic reporting

All comments have been added in detail to the last section.

Experimental design

All comments have been added in detail to the last section.

Validity of the findings

All comments have been added in detail to the last section.

Additional comments

Thank you for the revision. Both the responses to my referee comments and the changes in the paper are sufficient. I do not request any additional revisions. Best regards.

---

## Round 0.3 · accepted · Accept

I have checked the changes that you have made in response to the referee's comments and am satisfied that they are complete. You should ensure that the citation style matches the journal guidelines.